# LEETPROMPT: A CITIZEN SCIENCE PLATFORM FOR TESTING LLMS

## ABSTRACT

With the advent of pre-trained large language models (LLMs), natural language prompts are now becoming a de-facto method for interacting with language models. However, prompting as a technique is an esoteric art, involving cumbersome manual processes by individuals to search different strategies that make language models work for the intended task. We introduce LEETPROMPT, a citizen science platform that leverages on collective human creativity with prompting to solve reasoning questions across various domains. Users of LEETPROMPT attempt questions by writing prompts that solve all the hidden test cases. To measure the efficacy of LEETPROMPT, we conduct a study 10 questions across 5 domains (biology, physics, math, programming, and general knowledge) with 20 human subjects. We gather a total of 1173 GPT-4 prompts with the following observations: First, problems deemed unsolvable by question setters were successfully solved. Second, diverse prompting strategies were used by the different participants. Third, the more difficult problems also had a high number of prompt submissions enabling better debugging of the LLM behaviour for that problem. These observations support various downstream implications in robust approaches to prompt interpretability and model evaluation, high quality data collection, human-AI alignment and real-world usage of LLMs.
.

## 1 INTRODUCTION

Pre-trained large language models (LLMs) and the subsequent introduction of prompting, i.e. writing instructions to guide pre-trained models to perform tasks, has essentially ushered in a paradigm shift in machine learning research. Prompting as an input modality has influenced existing approaches in model interpretability (e.g., probing with prompts; Li et al., 2022), learning & alignment (e.g., RLHF; Ouyang et al., 2022), data requirements (e.g., instruction-tuning), and evaluation (e.g., prompts that get models to perform novel tasks; Bubeck et al., 2023), to initiating new avenues of research such as prompt optimization (Zhou et al., 2022; Zhang et al., 2022; Pryzant et al., 2023).

Despite the crucial role of prompting, the task of prompting continues to perplex the AI research community since obtaining insights through writing prompts is not only a cognitively demanding task, but can also quickly become tedious (Reynolds & McDonell, 2021; Jiang et al., 2022; Zamfirescu-Pereira et al., 2023). For example, if a researcher wishes to assess the capabilities of a language model, they must write prompts using diverse strategies one by one, unlike previously when they could benchmark the model straightaway against a dataset. Furthermore, while there are general techniques such as zero-shot encouragement (Kojima et al., 2022), chain-/tree-/program-of-thought (Wei et al., 2022; Yao et al., 2023; Chen et al., 2022), many strategies often vary between specific problems, their domains (Jia & Zhang, 2022), between each model and most importantly, may completely change over time. In such a complex space of prompts, conducting blind exhaustive search becomes excruciating, leading to erroneous assertions about the capabilities of language models. Such demanding nature of "prompt engineering" has also led to creation of new roles such as "prompt engineers" whose sole job is to discover prompts that steer the language model towards performing a particular task or analyzing the capabilities of language models.

Despite the trend towards optimizing prompts with automated methods, most of the already known popular strategies, such as chain-of-thought and GPT-4's ability to draw a unicorn in LaTeX, were

developed by substantial efforts and time devoted by individual researchers (Wei et al., 2022; Bubeck et al., 2023). Recognizing the importance of human creativity in prompt discovery, we introduce LEETPROMPT, a citizen science platform that enables researchers to upload questions, or models, in different domains and leverage the power of collective intelligence to solve these questions through prompting. Specifically, LEETPROMPT is a platform that hosts reasoning problems which regular people solve by writing (and engineering) prompts, while contributing towards a collectively knowledge of whether a problem is solvable and if so, how.

In order to evaluate the efficacy of LEETPROMPT as citizen-science infrastructure for testing LLMs, we conduct a controlled in-lab study with $N = 20$ human subjects. We invite $4$ initial users of the platform to select $10$ problems (out of a total of $101$ already on the platform) to be used for the study. Each problem consists of a question, a few examples, and corresponding $5$ hidden test cases. The $10$ problems equally span $5$ domains: biology, physics, math, programming, and general Knowledge. We chose these domains as representative of current testing and evaluation of LLMs (Hendrycks et al., 2020; Patel et al., 2021; Sap et al., 2019a). Participants have varying education background, experience with LLMs/programming, and demographics. The objective for the participant is to solve the given $10$ questions by writing prompts and pass all the hidden test cases within $1$ hour.

Through quantitative analysis and qualitative coding of 1173 prompts obtained in our controlled study, we have three key observations. First, problems that were deemed unsolvable by LEETPROMPT problem setters got solved over the many attempts by participants. Second, the more difficult problems received more number of attempts overall ensuring more fine-grained debugging data. Third, LEETPROMPT captures diverse approaches by participants eliciting natural human intuitions (or, preferences; Ziegler et al., 2019) towards solving the problem. Our study demonstrates the concept of LEETPROMPT and its implications on LLM research & development including "high quality" data collection, robust approaches to prompt interpretability and model evaluation, human-AI alignment and real-world usability of LLMs. LEETPROMPT is an instrument for testing, analyzing and evaluating LLMs (Olah et al., 2020), akin to microscopes that enable the study of microscopic organisms in biology.

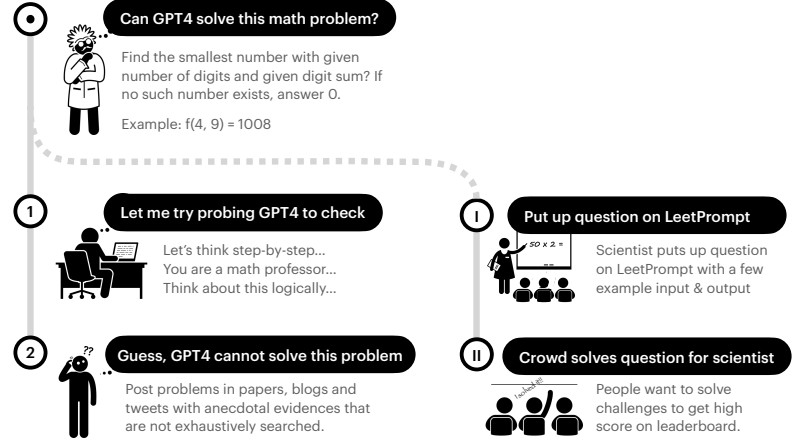

Figure 1: An example workflow of using LEETPROMPT for interpretability and evaluation of models. In this scenario, an AI researcher is curious about whether GPT4 can solve a particular question that they formulated. **1** & **2** demonstrate the traditional approach of the researcher independently attempting the question and drawing preliminary conclusions about the model. **I** & **II** demonstrate how the researcher posts up the question on leetprompt and leverages the power of collective intelligence to not only get the question solved but also explore all the different strategies used by others.

## 2 LEETPROMPT

We introduce LEETPROMPT: an online platform where users are presented with problems from multiple domains and tasked with writing prompts to get the LLM to solve the problems.

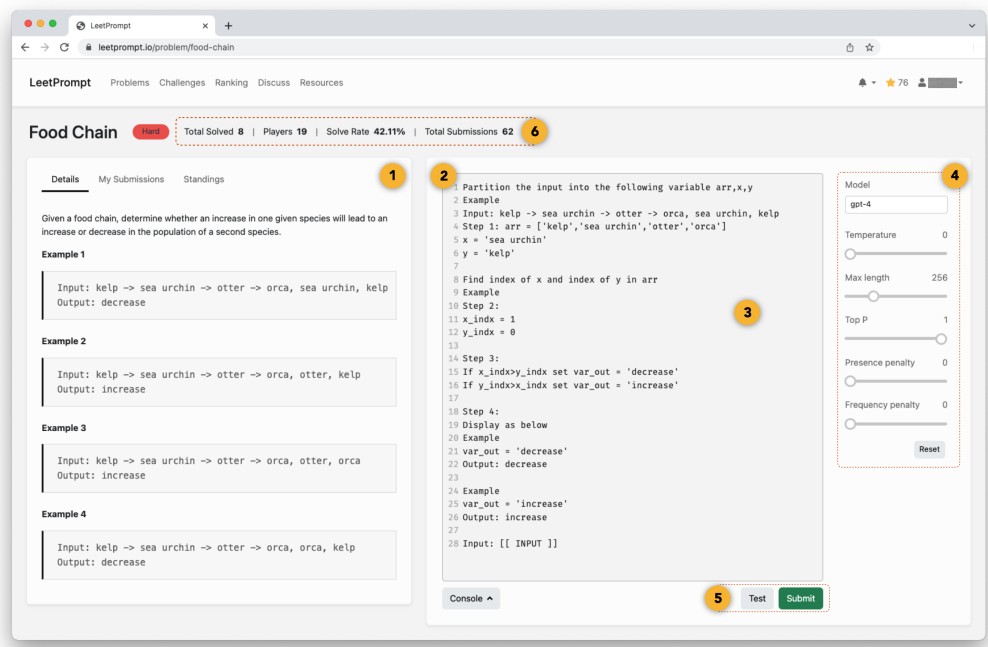

Figure 2: LEETPROMPT is a platform where users can explore the space of instructions to solve tasks with LLMs. **①** **Problem description.** This panel contains a description of the problem that needs to be solved. It often contains examples of plausible inputs and outputs. **②** **Interaction interface.** Here, users will write instructions, change model hyper-parameters, and evaluate their instructions against public as well as private test cases. **③** **Writing instructions.** Here lies the text interface where users write their instructions. The token [[ INPUT ]] identifies where test cases will insert inputs. Most problems already contain a starter instructions to help users get started. **④** **Model hyper-parameters.** This panel shows which model is being used and the corresponding modifiable model hyper-parameters. **⑤** **Test and submit.** Users can test their instructions against a custom input by clicking Test or submit their instructions to be evaluated against the blind test cases by clicking on Submit. **⑥** **Problem details and submissions.** Users can check overall problem statistics, load past submissions, and compare their performance against other users.

**Designing the LEETPROMPT workflow.** LEETPROMPT's core functionality is inspired by online code judging systems such as LeetCode, TopCoder, and CodeForces.[1] The platform contains a list of problems that users choose to tackle. Each problem also has a leaderboard indicating how many users have already attempted to tackle this problem, who achieved the best performance, and with how many submissions. Once the user chooses a problem, they are given the problem's description as well as a few examples of inputs and outputs outlining expected behavior. With this information, users must write prompts that steer a large language model toward solving the given problem. Users can also change the model's hyperparameters.

Once they are finished writing the prompt, users can test its utility against their own custom inputs. Custom inputs allows users to iteratively explore how their changes affect LLM behavior. Once satisfied, they can submit their prompt to be evaluated against our hidden test cases. These private test cases are never revealed to the user; instead, they receive a percentage score indicating how many of the private test cases passed. To discourage rapid-fire submission and hill-climbing behavior on the private test cases, the platform provides a time-out after 3 submissions for 5 minutes. LEETPROMPT also records how many times the user submits prompts, and uses that information to rearrange the leaderboard to tacitly encourage fewer submissions.

**Designing the LEETPROMPT's interface.** LEETPROMPT's interface is designed to be simple and promote an intuitive user experience (see Figure 2). We recruit UX designers to join our team

---

[1]leetcode.com, topcoder.com, codeforces.com

Table 1: User performance across the 10 questions in 5 domains. **Difficulty** of the question was pre-determined by the problem setters. **#Int** is the number of model interactions, **#Sub** is the number of instructions that were submitted, **Passed** is the number of passed test cases, and **Solved** is the percentage of participants who able to successfully solve the problem. **LD**, **SD**, and **AD** are the Lexical, Semantic, and Approach Diversity, respectively.

| Domain | Question | Difficulty | #Int | #Sub | Passed | Solved | LD | SD | AD |
|---|---|---|---|---|---|---|---|---|---|
| Biology | Water Potential | Easy | 30 | 37 | $4.72 \pm 0.83$ | 94% | 0.59 | 21 | 0.61 |
| | Food Chain | Hard | 76 | 75 | $3.07 \pm 1.47$ | 50% | 0.43 | 30 | 0.49 |
| Physics | Ideal Gas Law | Easy | 108 | 31 | $3.68 \pm 1.76$ | 88 % | 0.45 | 32 | 0.25 |
| | Resistance is Futile | Medium | 84 | 48 | $2.88 \pm 1.76$ | 50% | 0.43 | 34 | 0.39 |
| Math | Consecutive Integers | Easy | 67 | 25 | $4.36 \pm 1.66$ | 75% | 0.7 | 33 | **0.80** |
| | Digit Sum | Hard | 134 | 45 | $3.27 \pm 0.75$ | 6 % | 0.43 | 43 | 0.58 |
| Programming | Intersperse | Medium | 99 | 23 | $3.70 \pm 1.46$ | 63% | 0.59 | 29 | 0.74 |
| | Sort Numbers | Medium | 64 | 27 | $3.56 \pm 1.53$ | 56% | 0.49 | 34 | 0.68 |
| Knowledge | Beatles Title | Medium | 177 | 28 | $3.36 \pm 1.57$ | 38% | 0.55 | 50 | 0.44 |
| | Theory of Mind | Medium | 36 | 13 | $3.15 \pm 2.23$ | 44% | 0.43 | 15 | 0.60 |
| | **Average $\rightarrow$** | | 87.5 | 30.3 | $3.58 \pm 0.55$ | 56% | 0.49 | 32 | 0.57 |

and run studies using mockups of the interface to identify common pitfalls. We face the following design challenge: creating a platform that is intuitive for users who are not familiar with the process of writing prompts for LLMs. Drawing on design theory (Nielsen, 1995), the following principles are used: (1) Standardizing the interface to match with existing/popular LLM platforms and (2) employing recognition over recall so users know the full extent of the platform's affordances through a quick skim. The appendix contains older versions of the interface.

**Evaluating user-generated instructions.** Figure 2(C) contains a screenshot of a possible instruction that a user might write to solve a problem. Users are expected to place an `[[ INPUT ]]` token somewhere in their produced instruction. LEETPROMPT evaluates instructions using the following:

$$\text{accuracy} = 100 \times \frac{1}{N} \sum_{i=1}^{N} \mathbb{1}[y_i == LLM((\text{Instruction}; \text{[[ INPUT ]]} = x_i))] \qquad (1)$$

where $(x_i, y_i) \in [1, \ldots, N]$ are $N$ private test case (input, output) pairs. (Instruction; `[[ INPUT ]]` $= x_i$) replaces the input token in the instruction with test case input $x_i$. $\mathbb{1}[\cdot]$ is the indicator function. LEETPROMPT contains a set of common regexes to extract the desired output for the problem from the LLM's generation.

## 3 STUDY TO JUDGE EFFICACY OF LEETPROMPT

In the following section, we outline the study we conducted to determine the efficacy of LEET-PROMPT as a platform for the crowd to participate in by solving challenging questions while helping researchers gain insights into the working of language models.

**Domain selection.** LLMs are often evaluated on their ability to perform simple mathematical and commonsense reasoning (Hendrycks et al., 2020; Patel et al., 2021; Sap et al., 2019a; Tafjord et al., 2019), causal reasoning (Kıcıman et al., 2023) and theory of mind & social reasoning (Sap et al., 2022; Moghaddam & Honey, 2023; Sap et al., 2019b), basic sciences (Taylor et al., 2022), programming (Chen et al., 2021; Liu et al., 2023), and even law exams (Zhong et al., 2023; OpenAI, 2023). We choose a subset of these domains as the test bed for our user study: Biology, Physics, Math, Programming, and General Knowledge.

**Problem setting.** At present, LEETPROMPT consists a total of 101 problems uploaded by users of the platform. Each problem consists of a set of 5 private test cases, and is also assigned a difficulty by the problem uploader – Easy, Medium and Hard. We invite a small set of 4 initial users with prior experience interacting with and using LLMs to LEETPROMPT. We ask the users to try and solve

Table 2: Qualitative Coding of Instruction Strategies used. Code represents the main category, with several subcodes within each category.

| Strategy | Code | Sub Codes |
|---|---|---|
| Instruction Prompting | INST | EXP, SIM, INC, COMP |
| Examples Used | EX | ZERO, N-SHOT, ORDE |
| Chain-of-thought | COT | CONS, COMP, TEXT |
| Self reflection | SELF | ASK, TAUG |
| Structure | ST | BREAK, STEP, QUES, NONE |
| Code Used | CODE | PROG, PSEU |

Table 3: Correlation Coefficients between problem properties and difficulty, number of testcases passed and whether it was solved.

| | | Problem | | |
|---|---|---|---|---|
| | | **Difficulty** | **Pass Rate** | **Solve Rate** |
| **Prompt** | # Test Runs | 0.31 | -0.35 | -0.49 |
| | # Eval Runs | 0.58.. | -0.57.. | -0.46 |
| | LD | -0.34 | 0.71* | 0.32 |
| | SD | 0.37 | -0.32 | -0.15 |
| | AD | -0.02 | 0.34 | -0.03 |

each question and sample a total of 10 problems, 2 from each of the 5 aforementioned domains, for the purpose of user study. In total there are 3 easy problems, 5 medium problems and 2 hard problems. The 2 hard problems were chosen because the problem setters were not able to find a prompt that worked for all the test cases. Problems are randomly ordered for each user.

**Example problem.** Figure 2 lays out one of our Biology problems and a user-generated prompt. It presents the participant with a food chain in the form of $X \rightarrow Y$ implies that species $Y$ is at a higher tropic level and, therefore, eats species $X$. The user is also presented with two species sampled from that food chain. The problem asks the user to determine whether an increase in the population of the first species will lead to an increase or decrease in the second. Example 1 in the problem description presents the following food chain: kelp $\rightarrow$ sea urchin $\rightarrow$ otter $\rightarrow$ orca. The two species presented are "sea urchin" and "kelp". Since "sea urchins" eats kelp, an increase in the "sea urchin" population leads to a decrease in the "kelp" population. This is a hard problem.

The user-generated instruction uses pseudocode to inform the model that if the first species is at a higher trophic level than the second, the population of the second species will decrease, and vice versa. Even though this logic works for "sea urchin" and "kelp", this pseudocode is incorrect when the two species are "otter" and "kelp". Since there is a skip connection between "otter" and "kelp", an increase in "otters" will result in a decrease in "sea urchins", which will in turn result in an increase in "kelp".

**Study protocol.** We design a study to explore the utility of LEETPROMPT. The study begins by asking participants to sign a consent form that outlines that their interactions will be stored and used for this research contribution. Next, they are asked an initial survey to record their demographics, their background education, and their experience working with LLMs or programming more broadly. Next, they are shown instructions for using the interface and provided resources that they can reference for advice on good prompt design. After this, the participant is shown a demo of the interface, which highlights and explains the various panes in the interface. To help familiarize themselves with the interface, users are provided a sandbox interface with a toy starter problem that they can test the functionality afforded by the platform. Next, the participants are presented with the 10 sampled problems and asked to attempt solving each one. We set expectations that the study will likely take about 1 hour. The study ends with a final survey where we collect open-ended responses describing their experience.

**Participant selection.** All of our participants are recruited locally from the authors' city. We only include participants that speak English, can spend at least 1 hour working on problems, and submitted at least one prompt to each of the 10 problems. All participants receive a payment of $15 for their time with no bonuses. We provide no other incentives other than intrinsic motivation for subsequent attempts at designing better prompts for each problem. Regardless, we find that each user tried an average of 5.8 prompts per question.

**Measured variables.** To evaluate prompts, we use two primary objective performance measures: solvability and diversity. *Solvability* measures the percentage of test cases that pass when using a specific prompt. *Diversity* measures how creative people are in writing prompts. Diversity provides insights into the different approaches that a participant takes to solve each problem. It allows us to evaluate the claim that diverse prompts and ensembling leads to significant boosts in LLM performance (Yoran et al., 2023). Since diversity of prompts can be measured in multiple ways, we measure the following: (1) *Lexical Diversity (LD)*, *Semantic Diversity (SD)*, and *Approach Diversity (AD)*.

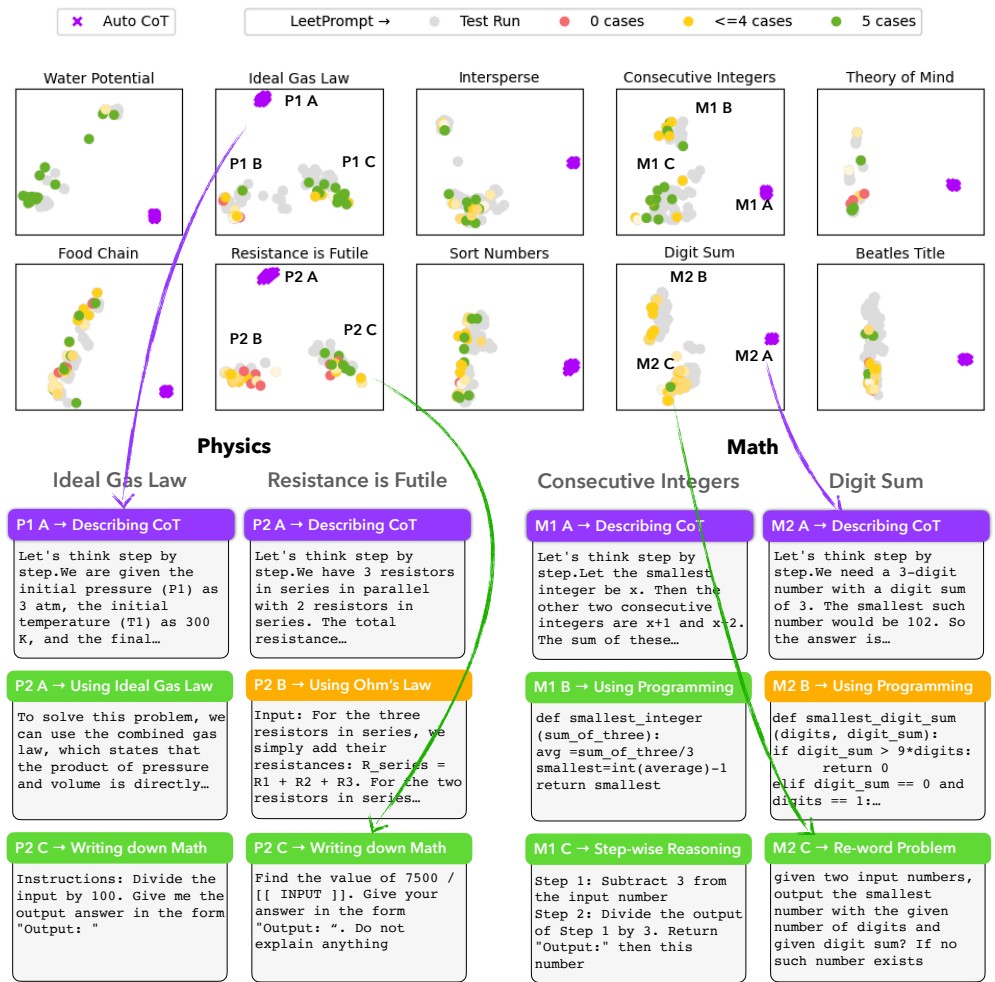

Figure 3: **Visualizing the space of Auto-CoT and LEETPROMPT Instructions**: 2D Principal Component Analysis (PCA) of embeddings of Auto-CoT and collected instructions from LEETPROMPT. Auto-CoT instructions are marked as purple X. LEETPROMPT instructions are marked as dots with colors representing its solvability: "Test" instructions are colored gray. For "Submitted" instructions, red color indicates that they failed all testcases, yellow indicates they passed 1-4 testcases, and green indicates that they passed 5 testcases. Instructions are specifically shown for two problems each from each domains to illustrate the different strategies used by participants and whether they were successful.

LD uses Repetition Rate (RR) (Cettolo et al., 2014; Bertoldi et al., 2013) to measure diversity in lexicons between a participant's prompt and the original wording used in the problem description. Specifically, LD = 1 − RR. LD is 0 if all lexicons appear more than once and 1 if all lexicon are new.

SD measures the diversity in strategies used by a user when tackling a specific problem. We use text+code embeddings (text-embedding-ada-002) to encode each prompt and visualize their first two principle components for qualitative visual analysis. For a quantitative analysis, SD is the variance for the first principal component.

AD measures the percentage of prompt strategies used, where strategies are qualitatively coded by 3 independent coders (see Table 2) for codes and appendix for full details). The qualitative codes reflect common prompt strategies in literature: prime the model to behave like a domain expert (Argyle et al., 2022), use example input/outputs (Wang et al., 2022a), use CoT prompting (Wei et al., 2022; Kojima et al., 2022; Wang et al., 2022b; Ye & Durrett, 2022; Turpin et al., 2023), use self-

help (Press et al., 2022; Zelikman et al., 2022) and the lexical structure (Fu et al., 2022), and use pseudo-code (Zhang et al., 2023a).

**Automatic prompting mechanisms.** We compare user-generated prompts with mechanisms used in existing benchmarks. Zero-shot (*0s*) uses the problem description and no other information. Zero-shot CoT (*CoT*) includes a phrase which requests the LLM to reason the steps aloud (Kojima et al., 2022). Few-shot variants (*N*-shot *Ns* where $N = 1, 2, 3, 4$) append $N$-example input/output pairs. We also test with advanced auto-prompting methods, such as *Auto-CoT* (Zhang et al., 2023b) which invokes multiple CoT reasoning steps along with 10 input/output examples, and Synthetic prompting (Shao et al., 2023) which tunes existing prompts.

**Models used.** LEETPROMPT supports any LLM that affords an API access to invoke the model. We allow participants to use any of the following models: GPT-4, GPT-3.5, and GPT-3. However, all participants chose to use GPT-4 for all their attempts. In the appendix, we retroactively evaluate their prompts on the other models as a point of comparison.

## 4 RESULTS

By analyzing the 1173 user-generated prompts and their feedback from the survey we find the following: To start, participants are able to generate prompts to solve all the problems, including the hard questions that the problem setters had deemed unlikely to be solved. Second, for more difficult questions, the users submit more prompts increasing the amount of data even though they might not be able to solve them. Third, there were many diverse prompt strategies that each participant used to solve these questions. Finally, the background of the participants had an impact on their ability to solve different problems.

**①  LEETPROMPT enables solving problems deemed unsolvable by problem setters** As shown in Table 1, all problems were solved by at least one participant. The problem setters' difficulty categorization is strongly correlated with how many test cases participants passed ($r(19) = .74, p = .01$) and what percentage of participants were able to solve all 5 private test cases ($r(19) = 0.84, p = .001$). For "Digit Sum", only one participant was able to solve the question. Surprisingly, the successful solution involved re-arrangement of examples and a specific re-wording of the question that improved the problem's clarity and thus made it easier for the model to understand and solve the problem. Only through collective action was LEETPROMPT able to identify this solution. Similarly, half the participants were able to solve "Food Chain". Surprisingly, one winning instruction was a logically incorrect reasoning step that somehow still passes the test cases (shown in Figure 2). This adds more support to concurrent work, which also finds that unfaithful reasoning steps improve LLM performance (Turpin et al., 2023).

Furthermore, none of the automatic mechanisms were able to find prompts to solve all the problems (Table 4) and the diversity metrics (LD and SD) are both significantly smaller. A low LD implies that these mechanisms do not deviate from the original problem description. We visualize the lack of diversity of Auto-CoT in Figure 3, which visualizes the first two principal components of the SD prompt embeddings for each problem.

**②  LEETPROMPT encourages more attempts on difficult problems.** From ① we also see that the problems labelled as 'Hard' have a higher number of submissions compared to other problems. For more challenging problems we see users persevere and actually generate much more data that can be further used later to aid in model interpretability. These prompt submissions can be studied to help debug the model's strengths and weaknesses for the type of problem.

**③  LEETPROMPT captures diverse prompting strategies used by participants.** Each column in Figure 3 is a different domain. Interestingly, the clusters appear to follow a similar pattern between the two rows, implying that people use similar strategies for problems within a given domain.

Not only did we have diverse prompting strategies, but also lexical diversity of prompts is positively correlated with higher pass rates ($p(19) = 0.71, p < 0.01$) (Table 3). This mirrors prior work (**?**) which found that having diverse reasoning allowed for better LLM performance. This result suggests that large LLM benchmarks should rethink the use of a small fixed set of prompts. From Table 4 the best automatic mechanism, Auto-CoT, has lower LD than LEETPROMPT participants (0.26 vs 0.49) and also passes fewer number of test cases (45 vs 50) which further proves this point.

Table 4: Comparison of existing prompting approaches to prompts collected from LEETPROMPT. **0s**: Zero-shot; **0s CoT**: Zero-shot Chain-of-Thought prompting; **1s, 2s, 3s, 4s**: 1,2,3,4 shot (or examples) prompting; **Auto-CoT**: Automatic Prompting Method; **Ours**: Prompts from LEETPROMPT. **P** denotes the maximum number of testcases that the the given method was able to pass. As Auto-CoT and Ours have multiple prompts per problem, we report **LD & CD** which are the lexical and content diversity of prompts for each problem. We do not report **AD** for Auto-CoT since it defaults to using CoT as the main strategy for solving the problem.

| Domain | Question | 0s | 0s COT | 1s | 2s | 3s | 4s | Auto-CoT | | | Ours | | |
|---|---|---|---|---|---|---|---|---|---|---|---|---|---|
| | | P | P | P | P | P | P | P | LD | SD | P | LD | SD |
| Biology | Water Potential | 2 | 2 | 3 | **5** | **5** | **5** | **5** | 0.17 | 2e-05 | **5** | 0.56 | 3e-02 |
| | Food Chain | 4 | 3 | 3 | 3 | 2 | 3 | **5** | 0.25 | 3e-05 | **5** | 0.48 | 6e-03 |
| Physics | Ideal Gas Law | **5** | 3 | 4 | 4 | 4 | **5** | 4 | 0.42 | 1e-04 | **5** | 0.38 | 7e-02 |
| | Resistance is Futile | 0 | 0 | 3 | 3 | 4 | 2 | 3 | 0.42 | 2e-04 | **5** | 0.43 | 8e-02 |
| Math | Consecutive Integers | **5** | 4 | **5** | 4 | 4 | **5** | **5** | 0.34 | 2e-05 | **5** | 0.68 | 5e-03 |
| | Digit Sum | 3 | 2 | 3 | 3 | 3 | 4 | **5** | 0.31 | 2e-05 | **5** | 0.46 | 5e-03 |
| Programming | Intersperse | **5** | **5** | **5** | **5** | **5** | **5** | 4 | 0.10 | 2e-05 | **5** | 0.51 | 8e-03 |
| | Sort Numbers | 0 | 0 | 4 | 4 | 3 | 3 | 4 | 0.37 | 9e-05 | **5** | 0.49 | 2e-03 |
| Knowledge | Beatles Title | **5** | **5** | 4 | 4 | 4 | **5** | **5** | 0.11 | 5e-05 | **5** | 0.47 | 3e-04 |
| | Theory of Mind | 0 | 0 | **5** | **5** | **5** | **5** | **5** | 0.11 | 3e-05 | **5** | 0.49 | 4e-04 |
| | | 29 | 24 | 39 | 40 | 39 | 42 | 45 | 0.26 | 6e-05 | **50** | **0.49** | **2e-02** |

Table 5: Pearson's correlation coefficient ($r$) between participant attributes (demographic, background, and experience) and the maximum number of test cases passed and the time taken for each problem. '..' indicates trending towards significance ($p < 0.1$) and '*' denotes significance ($p < 0.05$). **P** is the average number of testcases passed and **T** is the avg. time taken between first and last interaction with the problem .

| Domain → Participant ↓ | Biology | | Physics | | Math | | Program | | General | | Overall | |
|---|---|---|---|---|---|---|---|---|---|---|---|---|
| | P | T | P | T | P | T | P | T | P | T | P | T |
| **Demographic** | | | | | | | | | | | | |
| Age | 0.23 | -0.23 | -0.13 | 0.20 | 0.13 | 0.29 | 0.36 | 0.44.. | 0.13 | 0.23 | 0.21 | 0.32 |
| **Experience** | | | | | | | | | | | | |
| Biology | -0.03 | -0.11 | -0.32 | 0.03 | -0.41.. | -0.50* | -0.26 | 0.16 | -0.29 | -0.09 | -0.37 | -0.24 |
| Physics | 0.12 | -0.01 | 0.18 | 0.46.. | 0.04 | 0.03 | 0.21 | -0.08 | 0.31 | -0.23 | 0.25 | 0.02 |
| Math | 0.18 | 0.25 | 0.01 | 0.31 | 0.10 | 0.30 | 0.34 | 0.41.. | 0.25 | 0.27 | 0.26 | 0.54* |
| Trivia | -0.01 | 0.19 | 0.08 | -0.11 | 0.25 | -0.05 | 0.45.. | 0.34 | 0.22 | 0.01 | 0.30 | 0.11 |
| LLM | 0.09 | 0.51* | 0.14 | 0.22 | 0.43.. | 0.19 | 0.36 | 0.25 | 0.43.. | 0.42.. | 0.42.. | 0.59* |
| Prompting | 0.43.. | 0.13 | 0.21 | 0.54* | 0.35 | 0.16 | 0.35 | 0.06 | 0.25 | 0.35 | 0.43.. | 0.45.. |
| Program | -0.27 | 0.18 | -0.05 | -0.38 | 0.09 | -0.22 | 0.19 | 0.15 | 0.25 | 0.19 | 0.10 | -0.02 |

We also found that on average, natural language worked better than pseudocode. By qualitatively analyzing the clusters in Figure 3), we find that participants who used pseudocode prompts generally wrote longer prompts to account for all possible edge cases, similar to writing exceptions in software. Debugging these prompts was easier even though writing the complex prompt took longer. For those who wrote natural language prompts, their prompts were shorter, and more successful.

Explicit, detailed prompts do better but prompt structure can also affect performance. Participants learned a need for clarity and to "give more direct and structured commands" (P34) and be "extremely explicit and instructive" (P48) in order for the model to understand. Others noticed that relatively trivial changes led to different results in unpredictable ways, such as "'do not include any explanation and just provide the answer' changing the answer" (P48). Related, in "Beatles Title", we notice that the approaches did not cluster into easily separable clusters (Figure 3). This is because the question purely relied on factual knowledge and did not require explicit reasoning. In the absence of a strategy, we evaluated how minor changes in prompts might result in better performance, finding that generous use of line breaks between examples resulted in better performance. Our future work will evaluate the generality of this finding.

**④ LEETPROMPT allows observing participant behavior.** Participants with experience with LLMs, or an education in math spent significantly more time writing prompts ($p(19) = 0.59, p < 0.05$) and ($p(19) = 0.54, p < 0.05$). From Table 6, those with either experience with LLMs or prompting were trending towards significance for solving more problems ($p(19) = 0.42, p < 0.1$, $p(19) = 0.43, p < 0.1$). From Table 6, there is no strong correlation between the participant's domain knowledge with solving problems in that domain. Taken together, these two findings sug-

gest that knowing how to instruct the model can be more important than domain knowledge about the problem. Participants also demonstrated and self-reported a learning effect as they progressed through the study. Multiple participants mentioned that they began adopting specific strategies as they completed more questions. "Yes, as I went through the problems I learned how to ask the model questions so it could help me answer the question. I also learned how to provide reasoning for the examples." (P39) "I realized simplifying the problems greatly increased the reliablility of the model, so I tried to rephrase questions to be as simple as possible [for future problems]" (P42).

**5** **LEETPROMPT uncovers misalignment between users and models.** Participants reported a sense of confusion between their expectations versus how the model worked. Figure 3 shows visually that there exists entire clusters of instructions that do not solve the problem. For example, math-related strategies to solve the two math problems didn't work while programming-related strategies did. Participants complained that the model "would do math incorrectly" (P42). Similarly, using domain-specific information, such as using Ohm's Law to solve the "Resistance is Futile" Physics question failed while using math-related instructions sometimes succeeded. Even so, one participant exclaimed, "what was also strange was that the numerical answer the LLM provided could change based on seemingly trivial additions to the prompt: "I would perturb prompts in small ways that might unexpectedly change the output" (P37).

# 5 DISCUSSION & IMPLICATIONS

**Evaluation and Interpretability.** Benchmarking is the primary approach to evaluting machine learning models. But as prompt-based language models have become more prevalent, benchmarks are dealing with a number of problems. **1** Benchmarks such as BigBench (Ghazal et al., 2013), MMLU (Hendrycks et al., 2020), and HELM (Liang et al., 2022) provide standardized prompts that all models use for evaluation. However, a lack of performance on tasks cannot be directly linked to the model's incapacity to execute the task or its incapacity to comprehend the prompt. **2** Models trained on diverse data sources including data leaks and contamination of benchmarks, making many public benchmarks and evaluation on standardized tests questionable. (Jacovi et al., 2023) **3** Benchmarks are quickly outdated due to rapid progress in AI modelling, but without an insight of what is to be measured, leading to claims of "superhuman" performances. (Kiela et al., 2021) **4** Benchmarks work with datasets and hence at macro-level inhibiting micro-level analysis leading to papers that put out anecdotal evaluations (Bubeck et al., 2023; Qin et al., 2023). LEETPROMPT platform helps mitigate all 4 weaknesses of benchmarks. **1** demonstrates the power of crowd to perform exhaustive search to come up with novel solutions. Given the nature of LEETPROMPT, new and more challenging questions can be constantly uploaded on the platform preventing models learning from any data leaks. **3** shows the possibility of collecting large number of prompts that enable fine-grained analysis of which prompts work and which don't, and what tasks are solvable at large.

**Data Collection.** Data collection is the backbone of data-driven machine learning research. However, recent introduction of pre-trained language models is changing the landscape of data collection in several ways. For example, **1** Instruction data is becoming an increasingly valuable resource to the extent that researchers are distilling larger models for synthetic prompts (Wang et al., 2022b). **2** Collection of prompts and other data from traditional data collection systems like MTurk are deteriorating in quality due to linear rewards especially with with annotators using ChatGPT for annotating their responses (Veselovsky et al., 2023).LEETPROMPT shows the potential for data collection fulfilling these pain points of data collection by making it competitive, i.e. only paid for solving the question and also a platform for collecting prompt data in a natural setup.

**Human-AI alignment** With models being increasingly used for real-world tasks, human-AI alignment has taken a center stage (Ouyang et al., 2022). However, there are several hurdles on the way towards alignment. **1** A lack of human supervision as research is favoring synthetic content by distilling language models instead of getting human supervised data leading to data feedback loops (Taori & Hashimoto, 2023). **2** Lack of real-world usage of models makes it difficult to collect human preferences. LEETPROMPT enables collecting diverse but commonly approached strategies that do not fully work with the current models showing gaps in human-AI alignment along with the required fixes **5**.

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
