

Figure 4: *
Collected instructions with 32 participants

Figure 5: **Visualizing the space of Auto-CoT and LEETPROMPT Instructions**: 2D Principal Component Analysis (PCA) of embeddings of Auto-CoT and collected instructions from LEETPROMPT. Auto-CoT instructions are marked as purple X. LEETPROMPT instructions are marked as dots with colors representing its solvability: "Test" instructions are colored gray. For "Submitted" instructions, red color indicates that they failed all testcases, yellow indicates they passed 1-4 testcases, and green indicates that they passed 5 testcases. Instructions are specifically shown for two problems each from each domains to illustrate the different strategies used by participants and whether they were successful.

## A   SCALING UP USER STUDY

To see if our results generalize, we expand our user study to include 12 more participants, bringing us to a total of 32. The trends described in our main paper persist, with additional dimensions now statistical significant. In addition to the results already described in the main paper, Table 6 shows that prior language model and prompting experience significantly improve participants' performance on our problems **across all domains**. This new statistic adds further support to a point we made in our main paper: we noted that users self-reported a learning effect, where they improve their ability to write instructions as they tackle more problems.

The change in PCA clusters over time between the user study results reported in the main paper and the scaled-up user study results is depicted in Figure 5. Overall, cluster arrangements are similar across problems; however, human-generated instructions increasingly span across the first principal component for most problems, indicating increased diversity in human instructions. A greater number of instructions also provides a more reliable indication of each strategy's solvability.

Table 6: Pearson's correlation coefficient between participant attributes (demographic, background, and experience) and the maximum number of test cases passed and the time taken for each problem. '..' indicates trending towards significance ($p < 0.1$) and '*' denotes significance ($p < 0.05$). **Pass** is the average number of testcases passed and **Time** is the avg. time taken between first and last interaction with the problem .

| Domain → | Biology | | Physics | | Math | | Programming | | General | | Overall | |
|---|---|---|---|---|---|---|---|---|---|---|---|---|
| Participant ↓ | Pass | Time | Pass | Time | Pass | Time | Pass | Time | Pass | Time | Pass | Time |
| **Demographics** | | | | | | | | | | | | |
| Age | -0.36* | -0.09 | -0.57* | -0.11 | -0.27 | -0.04 | -0.38* | 0.08 | -0.16 | 0.15 | -0.44* | -0.02 |
| **Experience** | | | | | | | | | | | | |
| Biology | 0.13 | -0.06 | -0.04 | 0.35* | 0.02 | -0.33.. | -0.09 | 0.18 | 0.11 | 0.03 | 0.03 | 0.03 |
| Physics | -0.26 | -0.08 | -0.01 | 0.37* | 0.03 | -0.10 | -0.05 | -0.14 | 0.06 | -0.23 | -0.05 | -0.03 |
| Math | -0.03 | 0.14 | 0.08 | 0.28 | 0.24 | 0.20 | 0.18 | 0.26 | 0.23 | 0.06 | 0.19 | 0.30.. |
| Trivia | -0.01 | -0.11 | -0.00 | 0.14 | 0.37* | -0.20 | 0.30.. | 0.24 | 0.19 | -0.12 | 0.22 | -0.06 |
| **Experience** | | | | | | | | | | | | |
| LM | 0.44* | 0.48* | 0.22 | 0.28 | 0.54* | 0.37* | 0.49* | 0.16 | 0.55* | 0.39* | 0.58* | 0.60* |
| Prompting | 0.40* | 0.34.. | 0.20 | 0.45* | 0.38* | 0.39* | 0.35* | 0.00 | 0.39* | 0.33.. | 0.44* | 0.58* |
| Programming | -0.06 | 0.23 | 0.09 | 0.08 | 0.26 | 0.13 | 0.29 | 0.14 | 0.36* | 0.09 | 0.26 | 0.23 |

# B    PARTICIPANT DEMOGRAPHICS

Figures 6, 7 and 8 describe the participant demographics and experience as surveyed before the study, and their feedback after they finished the study.

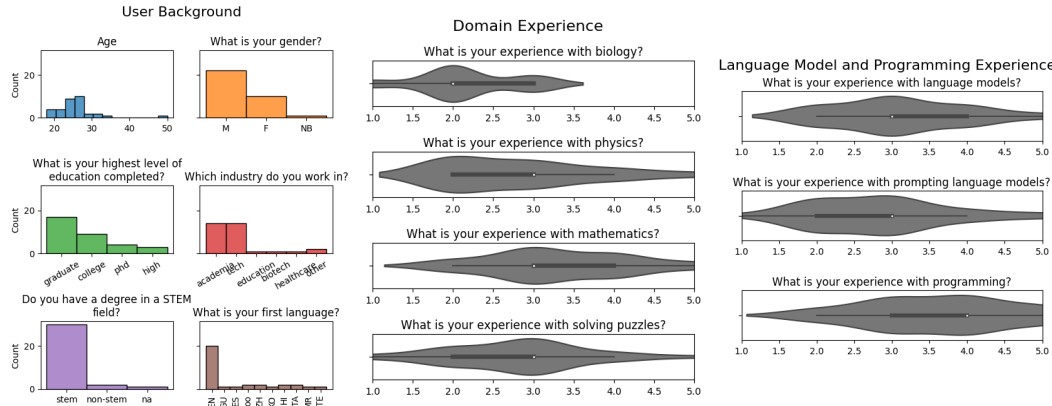

Figure 6: **Pre-study survey of user study participants** indicating their background (age, gender, highest level of education, industry, type of degree, and their first language), experience with different subjects (biology, physics, mathematics, and puzzles), and experience with using information technologies relevant to our study (language models, prompting, and programming)

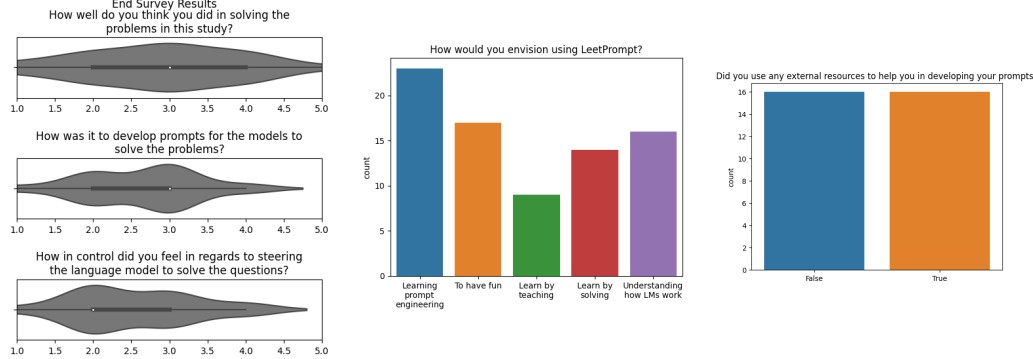

Figure 7: **Post-study survey of user study participants** describing their experience with solving problems, their perception of the platform and feeling of control with language models. Participants also report on whether they used external resources while solving problems and how they envision using leetprompt platform in the future.

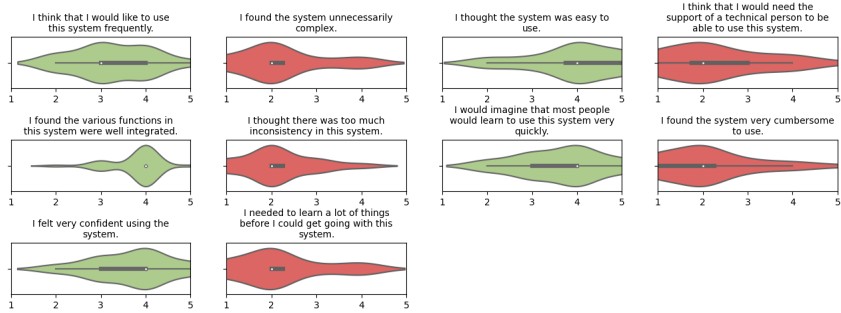

Figure 8: **System Usability Scale ?** used for measuring the usability of LEETPROMPT by study participants.

Table 7: *

Comparison of existing prompting approaches to prompts collected from LEETPROMPT using GPT-4, GPT-3.5-TURBO and GPT-3 as the language models. **0s**: Zero-shot; **0s CoT**: Zero-shot Chain-of-Thought prompting; **1s, 2s, 3s, 4s**: 1,2,3,4 shot (or examples) prompting; **Ours**: Prompts from LEETPROMPT. **P** denotes the maximum number of testcases that the the given method was able to pass.

### Table 8: GPT-4

| Domain | Question | 0s P | CoT P | 1s P | 2s P | 3s P | 4s P | Ours P |
|---|---|---|---|---|---|---|---|---|
| Biology | Water Potential | 7 | 7 | 8 | **15** | **15** | **15** | **15** |
| Biology | Food Chain | 9 | 8 | 9 | 6 | 6 | 8 | **15** |
| Physics | Ideal Gas Law | 12 | 7 | 10 | 12 | 12 | 13 | **15** |
| Physics | Resistance is Futile | 3 | 0 | 3 | 10 | 11 | 10 | **15** |
| Math | Consecutive Integers | 14 | 14 | 13 | 14 | 13 | 13 | **15** |
| Math | Digit Sum | 6 | 6 | 9 | 8 | 8 | 9 | **15** |
| Programming | Intersperse | 13 | 13 | 14 | 14 | 14 | 14 | **15** |
| Programming | Sort Numbers | 0 | 0 | 13 | 13 | 13 | 14 | **15** |
| Knowledge | Beatles Title | 14 | 14 | 11 | 14 | 14 | 14 | **15** |
| Knowledge | Theory of Mind | 1 | 0 | **15** | **15** | 14 | **15** | **15** |
| | | 79 | 69 | 105 | 121 | 120 | 125 | **150** |

### Table 9: GPT-3.5-TURBO

| Domain | Question | 0s P | CoT P | 1s P | 2s P | 3s P | 4s P | Ours P |
|---|---|---|---|---|---|---|---|---|
| Biology | Water Potential | 3 | 9 | 12 | 14 | 14 | **15** | **15** |
| Biology | Food Chain | 0 | 1 | 9 | 6 | 6 | 8 | **15** |
| Physics | Ideal Gas Law | 0 | 3 | 4 | 10 | 13 | 11 | **15** |
| Physics | Resistance is Futile | 0 | 0 | 0 | 2 | 2 | 2 | **15** |
| Math | Consecutive Integers | 10 | 13 | 14 | 11 | 12 | 8 | **15** |
| Math | Digit Sum | 7 | 3 | 5 | 6 | 6 | 5 | **14** |
| Programming | Intersperse | 6 | 2 | 1 | 11 | 11 | 12 | **15** |
| Programming | Sort Numbers | 1 | 0 | 11 | 12 | 12 | 13 | **15** |
| Knowledge | Beatles Title | 11 | 10 | 11 | 12 | 12 | 11 | **15** |
| Knowledge | Theory of Mind | 2 | 1 | 10 | 8 | 7 | 8 | **14** |
| | | 43 | 33 | 73 | 88 | 92 | 94 | **148** |

### Table 10: GPT-3

| Domain | Question | 0s P | CoT P | 1s P | 2s P | 3s P | 4s P | Ours P |
|---|---|---|---|---|---|---|---|---|
| Biology | Water Potential | 6 | 0 | 8 | 10 | 12 | 14 | **15** |
| Biology | Food Chain | 9 | 8 | 8 | 10 | 9 | 11 | **15** |
| Physics | Ideal Gas Law | 0 | 0 | 6 | 7 | 8 | 8 | **15** |
| Physics | Resistance is Futile | 0 | 0 | 0 | 3 | 4 | 4 | **15** |
| Math | Consecutive Integers | 7 | 7 | 9 | 4 | 4 | 6 | **15** |
| Math | Digit Sum | 4 | 3 | 4 | 3 | 3 | 5 | **12** |
| Programming | Intersperse | 0 | 0 | 4 | 9 | 10 | 10 | **13** |
| Programming | Sort Numbers | 0 | 0 | 6 | 5 | 2 | 9 | **15** |
| Knowledge | Beatles Title | 9 | 11 | 10 | 9 | 7 | 8 | **15** |
| Knowledge | Theory of Mind | 0 | 0 | 9 | 10 | 10 | 10 | **13** |
| | | 32 | 38 | 68 | 74 | 71 | 86 | **143** |

## C  EVALUATING OTHER MODELS

In this section, we evaluate how the human-generated instructions work across other LLMs. All the instructions were generated using GPT-4 interactions. Here, we test if those same instructions work on GPT-3 and GPT-3.5. We also add 10 new internal test cases along with the 5 externally generated test cases reported in the main paper.

Table 10 shows the performance of the instructions using GPT-3 (text-davinci-003) as the language model. We also show the results on all test cases (5 external + 10 internal). The instructions submitted by the study participants passed 143 out of 150 test cases which surpasses all the automatic strategies. The best performing automatic method, 4-shot, passes only 86 test cases, which accounts for only 58% of the test cases that human instructions succeed on.

Table 9 shows the performance of the instructions using GPT-3.5 as the language model. The instructions submitted by the study participants passed 148 out of 150 test cases which surpasses all

Table 11: **Summary of Problems** given in the user study, with an input / output example.

| | Question | Description | Example |
|---|---|---|---|
| **Biology** | Water potential | *Given the sucrose concentration of an animal cell in a solution, determine whether it will shrink or expand* | `[] INPUT: 11`
`OUTPUT: expand` |
| | Food chain | *Given a food chain, determine whether an increase in one given species will lead to an increase or decrease in the population of a second species.* | `[] INPUT: kelp -> sea urchin -> otter -> orca, otter, kelp`
`OUTPUT: increase` |
| **Physics** | Ideal gas law | *Deriving final pressure with constant volume using the ideal gas law PV = NRT* | `[] INPUT: 400`
`OUTPUT: 4` |
| | Resistance is futile | *Determining current for an electrical circuit given a voltage and resistance.* | `[] INPUT: 100`
`OUTPUT: 75` |
| **Math** | Consecutive integers | *Given a sum of three consecutive integers, find the smallest integer.* | `[] INPUT: 63`
`OUTPUT: 20` |
| | Digit sum | *Given two input numbers, output the smallest number with the given number of digits and given digit sum.* | `[] INPUT: 4, 9`
`OUTPUT: 1008` |
| **Programming** | Intersperse | *Insert a number 'delimeter' between every two consecutive elements of input list 'numbers'* | `[] INPUT: [1, 2, 3], 4`
`OUTPUT: [1, 4, 2, 4, 3]` |
| | Sort numbers | *Given an input of space-delimited string of numberals from 'zero' to 'nine', sort them from smallest to largest* | `[] INPUT: 'three one five'`
`OUTPUT: 'one three five'` |
| **General Knowledge** | Beatles | *Given a funny phrase, how many Beatles song titles are in it?* | `[] INPUT: Yesterday I toured a yellow submarine.`
`OUTPUT: 2` |
| | Housesitting | *Theory-of-mind: Bob has to go on a trip for his job. He has to leave his house - and his dog - for a week while he's on the trip. He is having his friend Anna take care of the house and the dog, Fido, while he's away.... Anna completes the given action. When Bob comes back, where will he look for the given item? Will he find it?* | `[] INPUT: Anna takes out Fido's treat bag to feed him after he sits on command. She leaves the bag on the counter. Bob comes back and Fido welcomes him like a very good boy! Bob wants to feed him a treat.`
`OUTPUT: G, no` |

the baseline prompting strategies by a significant margin. Again, the best performing automatic method, 4-shot, passes only 94 test cases, which accounts for only 63% of the test cases that human instructions succeed on.

Finally, table 8 shows the performance of the instructions using GPT-4 as the language model. The number of test cases passed is higher for all instruction strategies when using GPT-3 and GPT-3.5. The instructions submitted by the participants pass all 150 test cases, while 4-shot prompting passes 125 test cases which is the highest amongst the 3 models. Therefore, automatic methods using GPT-4 only pass 83% of all the successful human generated instructions.

Overall, we can see that despite the model we use, the human-generated instructions consistently outperform the automatic strategies. Even on less powerful models like GPT-3 and GPT-3.5 the human instructions pass more than 95% of the test cases, demonstrating the importance of studying LLM capabilities with human interactions.

## D    DIVERSITY OF PROBLEMS

We provide (in Table 11) a summary of all 10 problems with the description of each problem and some example input-output pairs that participants were shown as part of the problem description. Below, we list the reasons for choosing each of these problems to include in our user study.

**Water potential.**    We chose "Water potential" because it is a very simple problem in biology. We wanted to mix simple and difficult problems to see how the task's complexity influenced how users developed instructions. Most users noticed the greater than/less than relationship with 10, but even if they didn't, copying and pasting the problem statement and providing a few examples worked fairly well.

**Food chain.**    "Food chain" is a more difficult biology problem that the problem setters were unable to solve. It required much more complex logic that differed depending on the relative position of the two species in the food chain. Some participants asked the model how to solve it and gave that back to the model, which worked fairly well, while others gave incorrect logic, which worked in a few cases despite being factually incorrect.

**Ideal gas law.**    We chose "Ideal gas law" because it is one of the most fundamental equations in physics. Users didn't have to do much except apply the equation, which they could easily reduce to a simple division by 100, as many quickly realized. With this problem, however, a copy-paste strategy or even leaving out the explanation and asking the model to detect a pattern worked extremely well.

**Resistance is futile.**    "Resistance is futile" necessitated more logic in calculating the total resistance of the circuit prior to applying Ohm's Law. Despite the fact that Ohm's Law is fairly simple, the equations for calculating total resistance were too complex for the language model, and some participants found the text description of the circuit difficult to interpret. No one was able to solve this without simplifying the formula in our first round of user studies, but one participant in the second round was able to solve it with a vague prompt that did not include a formula and some examples.

**Consecutive integers.**    "Consecutive integers" is the simpler of the two math problems. It is an elementary school level problem that the language model easily understands. Users who simplified the formula were successful, but it was also possible to solve the problem by simply pasting or rewording the prompt and providing examples.

**Digit sum.**    "Digit sum" is an intriguing problem because it is very simple for a human to solve and is also considered an elementary school level problem. However, the logic is much more difficult to explain to the language model because the model isn't as strong in math and, in many cases, doesn't know what a "digit" is. Participants were surprised by the resulting outputs of their test inputs, and found it difficult to understand why the model produced those results. Even though it was very simple to solve manually, the problem setting team was unable to solve it using the language model. In this problem, only two instructions worked; both used a rewording of the question and two examples that were the same and in the same order, and neither example was an edge case example.

**Intersperse.**    "Intersperse" is a simple programming problem that participants with limited programming experience could understand. The problem setting team derived this problem from an open dataset Chen et al. (2021) rather than creating it. Some participants were surprised by the output because it provided a code to solve the problem rather than the solution, with one participant even adding "Please no code" to their prompt.

**Sort numbers.**    "Sort numbers" is another relatively simple programming problem, a simple array sort with English numbers rather than numerals. A version of this problem is also used in the open dataset Chen et al. (2021). The majority of participants explicitly converted between the text versions of the numbers and the numerals and created an array, while some successfully sorted the words directly.

**Beatles.**    We selected the "Beatles problem" because it is more concerned with general knowledge and text processing than with mathematical formulas or programming logic. To identify Beatles songs, the model needed to recognize them and be able to parse an input string. The model had trouble recognizing the song "Rain" in one of the example problems, which stumped participants who were trying to pass every example case before submitting, but because it was not used in the test cases, those who submitted anyways passed. The language model also counted additional titles that were semantically similar to phrases in the passage but were not direct substrings.

**Housesitting.**    "Housesitting" is a theory of mind problem. The problem-solving team wanted to know if the language model could perform well in theory-of-mind tasks. Participants were required to explain its lengthy description to the model. However, once participants provided all of the scenario descriptions, the language model was mostly successful in solving the problem. However, because of time constraints and a general dislike of reading long problem descriptions before being able to solve the problem, some participants were discouraged from even attempting the problem. The standard strategy participants employed, which was mostly successful, was to copy and paste the scenario description and insert some examples.

# E    QUALITATIVE CODING

To gauge the diversity in responses, we implemented qualitative coding. This method is typically used in social sciences to categorize and analyze qualitative data - in this case, submissions made in response to instructions. In this process, we assign codes, or specific labels, to different aspects of the data in order to classify it in a meaningful way.

Here are the codes that we utilize:

**Instruction prompting (`INST`):**    This coding category pertains to strategies involving direct instructions. These are the most common methods employed by participants. This might involve:

- `INST-SIM`: Simplifying the problem to make it more understandable.
- `INST-EXP`: Asking the model to emulate a third party, like an expert or a crowd, to generate a response. Argyle et al. (2022)
- `INST-INC`: Includes instructions that may not be factually correct but can still assist the language model in problem-solving. Turpin et al. (2023)

**Examples (`EX`):**    This category involves providing examples, which have been observed to enhance the model's problem-solving capacity.

- `EX-ZERO`: No examples are provided, leaving the model to interpret potential inputs and outputs. Kojima et al. (2022)
- `EX-N-SHOT`: Few examples are given, providing some clues to the model.
- `ORDER`: The solution utilized an unusual sequence in which examples are presented. Wang et al. (2022a)

**Chain of thought (`COT`):**    Encouraging the model to break down the explanation into steps or providing step-by-step problem-solving instructions can enhance the model's performance.

- `COT-CONS`: Changing the decoding strategy to promote diverse sampling with self-consistency. Wang et al. (2022b)
- `COT-COMP`: Using complex reasoning steps to assist the model. Fu et al. (2022)
- `COT-TEXT`: Indicates that a chain of thought approach doesn't substantially help with text-based problems. Ye & Durrett (2022)

**Structure (`ST`)**    The way the prompt text is formatted can influence the model's performance.

- `ST-NONE`: Continuously formatted instructions without line breaks. These appeared to be less effective.
- `ST-BREAK`: Breaking the prompt into multiple lines. The most commonly employed formatting strategy.
- `ST-STEP`: Structuring the prompt with steps or bullet points. This was shown to enhance prompt clarity.
- `ST-QUES`: Using "Q:" instead of "Question:". We observed this to be effective in certain cases. Fu et al. (2022)

**Writing code (`CODE`):**    Writing code (CODE): This category pertains to responses involving coding. Zhang et al. (2023a)

- `CODE-PSEU`: Writing solutions in pseudocode format.
- `CODE-PROG`: Writing actual functions in a programming language or asking the model to generate code.

**Asking the model for help (`SELF`):** In some instances, even if the model doesn't have a final solution, it can still provide helpful input as an intermediate step toward helping the user create a solution.

- `SELF-ASK`: Repeatedly asking the model for help can yield beneficial results. Press et al. (2022) (SELF-ASK)
- `SELF-TAUG`: When prompted with a few rationale examples as a self-taught-reasoner, the model generates rationales to answer many questions. Zelikman et al. (2022)

**Strategies for future studies** We found more strategies that we didn't code for in our responses, but we expect to appear in future studies with the platform.

- Social engineering the model such as giving it confidence or threatening it **?**. We did not see any of our participants do this, but they may attempt in future studies, as more awareness of this technique percolates to the public.
- Generate programs as the intermediate reasoning steps, but offloads the solution step to a runtime such as a Python interpreter **?**. We did not have any integrations with any code interpreters, but if we were to build such a feature into LEETPROMPT this would be useful coding.
- 'Program of Thoughts' (PoT) uses language models to express the reasoning process as a program and executes the code on an external computer. Chen et al. (2022). We did not have any integrations that run code, but if we were to build this feature into LEETPROMPT, this would be a useful coding.

## F DIVERSITY OF INSTRUCTIONS

In this section, we visualize some example human-generated instructions from our user study, how many test cases the instruction passed, and how we codified the instruction strategy with an explanation for our code. The following are a few sample instructions submitted for the problem "Resistance is Futile":

**Human-generated instruction:**

```
Find the value of 7500 / [[ INPUT ]].
Give your answer in the form "Output: "
```

**Number of test cases passed:** 5/5

**Explanation:** This solution presents a direct instruction approach, incorporating a simplified formula without providing any examples. Given its simplicity, there's no break line, list or question/answer structuring within the prompt itself.

**Coding:** `INST-SIM, EX-ZERO, ST-NONE`

**Human-generated instruction:**

```
For the following resistance , output the total current of the circuit
Input: 100
Output: 75
Input: 250
Output: 30
Input: 300
Output: 25
Input: [[ INPUT ]]
```

**Number of test cases passed:** 5/5

**Explanation:** This solution uses direct instruction, void of any simplified formulas. It reconstructs the question into a more straightforward form, stripping away details that might seem essential to a human, such as the configuration of the resistors or the battery's voltage. Nonetheless, the model manages to infer an answer from the three examples provided, which are separated by new lines. Interestingly, the sequence of these examples deviates from the original list.

**Coding:** `INST, EX-3-SHOT, ST-BREAK, ORDER`

**Human-generated instruction:** `Tell language model how to solve the`
```
problem here
You are building a device to resist the Borg.  In order to do
this, you need to connect some resistors in a circuit with a 9
V battery.  You have 5 resistors of a given resistance (in ohms).
You plan to connect 3 of them in series in parallel with two of
them in series.  For the three resistors in series, we simply add
their resistances:
Rseries = R1 + R2 + R3.  For the two resistors in series, we add
their
resistances:  Rparallel = R4 + R5.  Then we can calculate the
equivalent
resistance of the two sets of resistors in parallel:  1/Rparallel
=
1/Rseries + 1/Rparallel.Finally, we can use Ohm's Law to calculate
the
total current in the circuit:  I = V/Rtotal.  What is the total
current in
milliamperes of the circuit?

Input:  100
Output:  75
Input:  300
Output:  25
Input:  750
Output:  10
Input:  250
Output:  30
Input:  [[ INPUT ]]
```

**Number of test cases passed:** 4/5

**Explanation:** In this submission, the participant restates the problem statement while incorporating additional instructions and logical steps needed to calculate the current, a strategy characteristic of a 'chain of thought' prompt. The participant also provides four example cases, adhering to the order in which they were presented on the platform.

**Coding:** `INST, COT, EX-4-SHOT, ST-BREAK`

**Human-generated instruction:** `Here is a electrical circuit question I am`
`trying to solve:`
`What is the total current in milliamperes of the circuit?`

`Based on ohm's law, current is voltage devide by resistance,`
`the circuit has a voltage source of 9 volts, I will provide a`
`number,`
`the resistance of whole circuit will be 1.2 times the number I`
`provided,`
`can you calcuate the current for me?`

`The following are some example inputs and outputs:`
`Input:  100`
`Output:  75`
`Input:  300`
`Output:  25`
`Input:  750`
`Output:  10`

`Please note that the output of the current is milliampere, just`
`output a number without unit`

`Input:  [[ INPUT ]]`

**Number of test cases passed:** 5/5

**Explanation:** This solution employs a 'chain of thought' strategy, as it involves the participant explaining the calculation process for the current using a somewhat simplified formula. It also demonstrates the participant's effort to correct the model's behavior by explicitly stating that the output should be in milliamperes at the conclusion of the prompt.

**Coding:** `INST,COT,EX-3-SHOT`

**Human-generated instruction:**

```python
def total_current(resistance, battery_voltage):
    # Calculate the total resistance of the circuit
    series_resistance = 3 * resistance
    parallel_resistance = resistance + resistance
    total_resistance = series_resistance + (1 / parallel_resistance)

    # Calculate the total current in milliamperes using Ohm's law
    total_current = battery_voltage / total_resistance
    total_current_milliamps = total_current * 1000

    return total_current_milliamps

# Example usage
resistance = 100 # ohms
battery_voltage = 9 # volts
total_current_milliamps = total_current(resistance, battery_voltage)
print(""The total current in milliamperes is:"", total_current_milliamps)

Input: Any number
Output: total_current_milliamps
Input: [[ INPUT ]]
```

**Number of test cases passed:** 0/5

**Explanation:** This submission used Python code to create a solution that calculated the total current in milliamperes of the circuit. This required a detailed explanation of the current calculation method, which suggests the use of a 'chain of thought' prompt strategy. However, the method used was incorrect. Direct instructions were also a feature of this approach, as they guided the model to print a specific statement. Interestingly, this participant did not provide any example input-output pairs. The structure of the solution was enhanced by placing instructions on separate lines.

**Coding:** INST-INC, COT, CODE-PROG, EX-ZERO, ST-BREAK

**Human-generated instruction:** `You are an expert electrictian.`
`I give you 5 resistors all of the same resistance as input.  In`
`your circuit is a 9 V battery.`
`You have connected 3 resistors in series, which is in parallel`
`with 2 other resistors that are in series.  What is the total`
`current in milliamperes of the circuit?`

`Examples:`
`Input:  100`
`Output:  75`

`Input:  300`
`Output:  25`

`Now it's your turn:`
`Input:  [[ INPUT ]]`

**Number of test cases passed:** 3/5

**Explanation:** This participant has asked the model to simulate an expert electrician when giving instructions. They have also provided two examples which are separated by new lines.

**Coding:** `INST-EXP, EX-2-SHOT, ST-BREAK`

Participants demonstrated a diverse range of strategies in attempting to solve the problem, illustrating the rich array of thought processes that emerges when different individuals tackle the same challenge. However, the effectiveness of these strategies varied. Writing code and instructing the model to impersonate an expert was less successful for this problem. Alternatively, strategies that simplified the problem were more effective. This was seen in both the transformation of the problem into a straightforward formula, and the removal of seemingly crucial problem parameters. It turns out that, in many cases, these elements were important from a human perspective but not necessary for the language model to infer a solution. In conclusion, strategies that focused on distilling the problem to its core components were typically more successful.

# G USER INTERFACE DESIGN

The platform underwent multiple iterations to improve user experience and testing.

**Initial design.** Overall, the initial UI lacked clarity in presenting primary actions and information. We brought in UI/UX engineers onto our team to improve the design. With them, we identified several areas for improvement. We describe these improvements in Figure 9

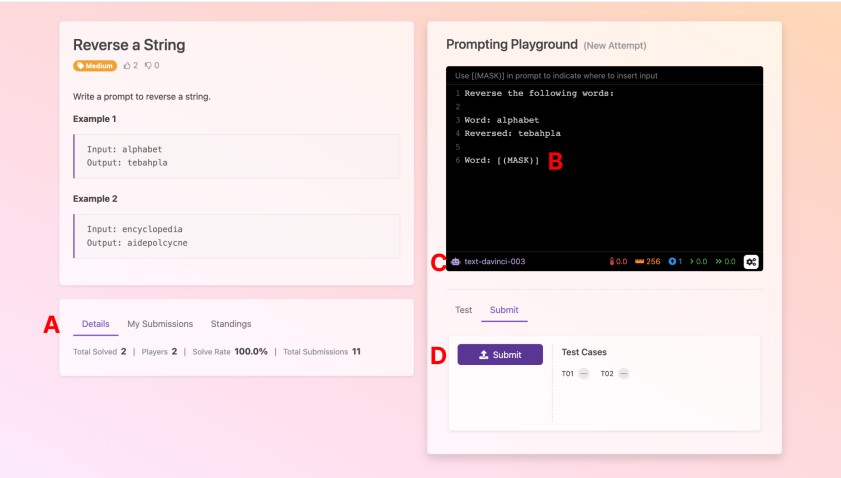

Figure 9: **Initial design details.** (A) The area containing "Details," "My Submissions," and "Standings" was located underneath problem description. If the problem description is too long, user may not see this information. (B) "Mask" represented the input that users could manipulate in order to test instructions, but the term was confusing to many users. (C) Unclear that user can switch to different models or change parameters. Icons do not clearly indicate what types of parameters can be changed. (D) Buttons to test and submit instructions were located under respective tabs. Users found this frustrating as testing and submission required an extra click into the tab before clicking on respective button.

**Design iterations.** Our team iterated through different versions in order to address the areas identified above. In Figures 10, 11, you can see two different designs with features used in the final user study design.

**Final UI design.** The final user study incorporated elements from previous iterations (see Figures 13, 14, 15). To improve usability, we included a tour of the interface (Figure 12), a problem navigation pane(Figure 13), and a test feedback UI change (Figure 16).

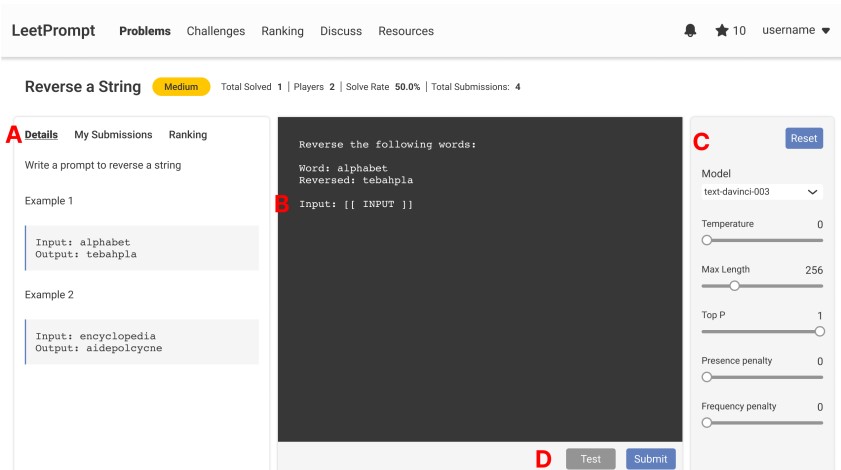

Figure 10: **Iteration 1.** (A) Problem description, details, user submissions, and user ranking were consolidated into one portion so users can easily scan for information available to them. The information previously found under "Details" was placed next to the problem name. "Details" in this iteration shows the problem description and examples. (B) The word "MASK" is replaced with the word "INPUT" so users understand that this text is to include their manipulated input. (C) Adjustment for models and parameters made more visible. (D) Test and submit buttons are visible.

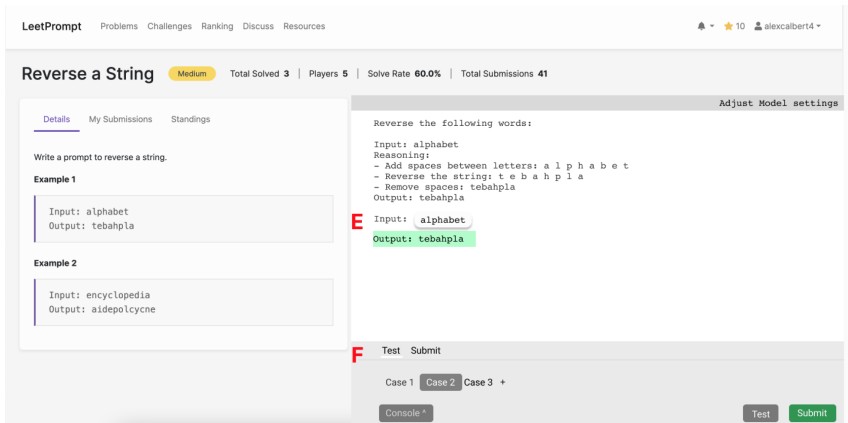

Figure 11: **Iteration 2.** In this second iteration, we explored to idea of (E) consolidating inputs and outputs into one area. Inputs would be highlighted based on the test case selected. And outputs would be highlighted in color. (F) Console expands and collapses so users have maximum area to work on instructions.

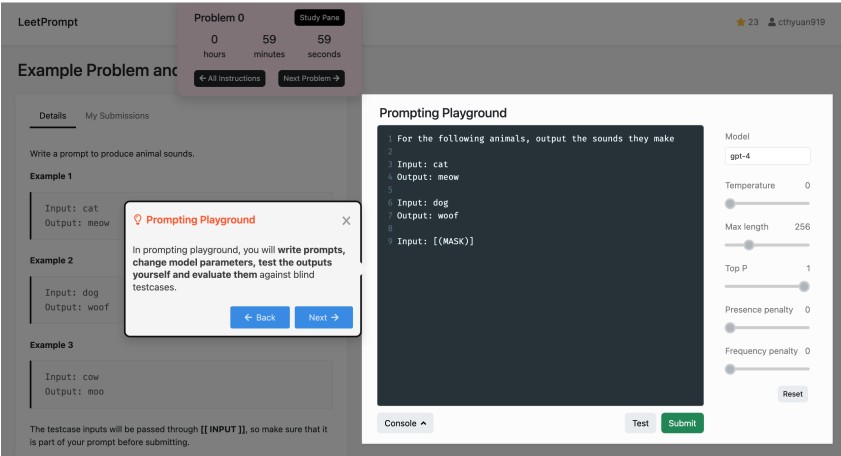

Figure 12: **Example problem and tour of the interface.** Before starting the user study, participants are given an example problem. A walk-through with tooltips introduces each section of the interface.

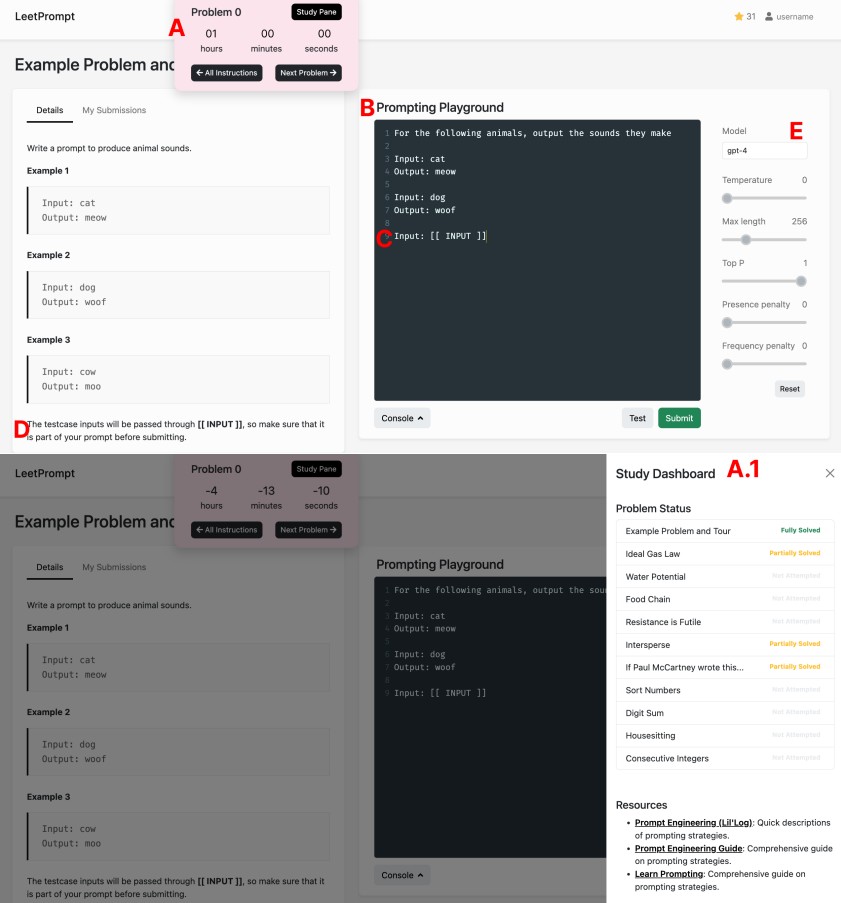

Figure 13: **User study interface.** For the user study, we adapted the interface to provide additional information to the study participants. For this, we made the following changes as marked: (A) Modal overlay provides problem navigation and shows time remaining in the study. The button "Study Pane" leads to a study dashboard side sheet (A.1) where participants can navigate to other problems and see the status of each ("Fully Solved," "Partially Solved," "Unsolved," and "Not Attempted). (B) Explicitly stating the name of this section. (C) & (D) Clarification of relationship between input and test cases. (E) Model and parameters adjustments are visible but frozen for the purposes of this study. This is explained in the initial walk-through.

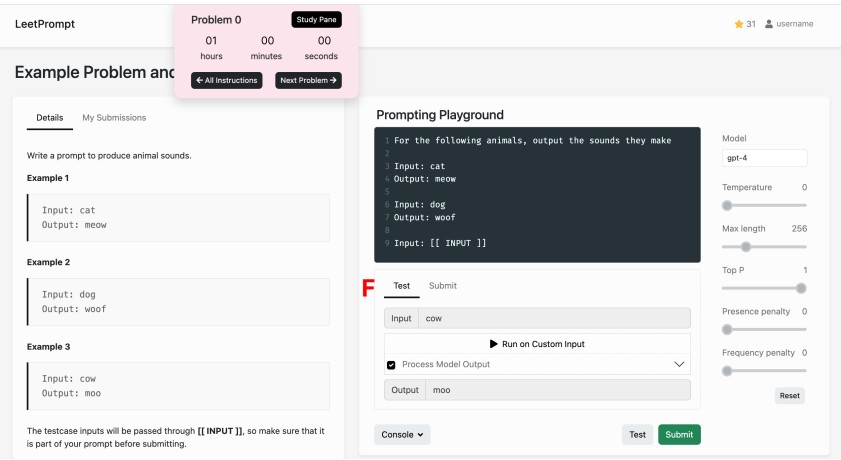

Figure 14: **Testing functionality.** The participants can test their instructions with their own custom inputs in the test console. The specific functionalities are as follows: (F) Participant's text input was divided into three sections. The top screen allows editing for instructions. Bottom section shows area to enter and edit input. Area that shows output of model is below. Participant can either click on "Run on Custom Input" or the "Test" button to test instructions on a particular input.

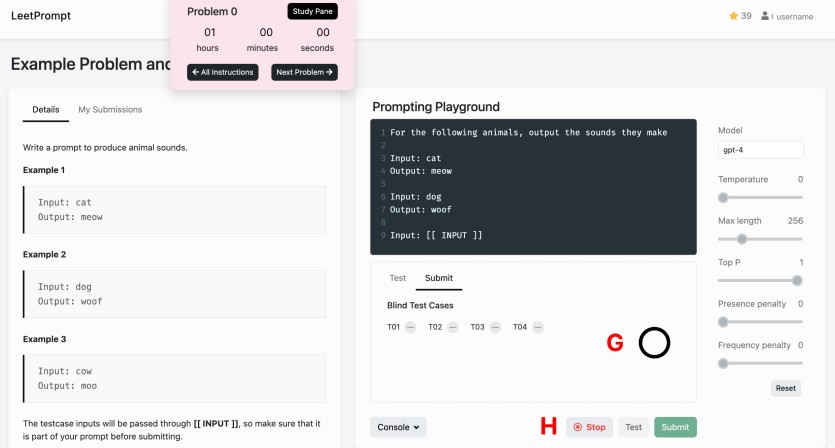

Figure 15: **Submit functionality.** After the participants are done testing their instructions, they can submit their prompt for evaluation against the blind testcases. When the "Submit" button is clicked, participants are taken to the submit tab. (G) Loading animation was added to indicate progress. (H) Stop button added for better user control.

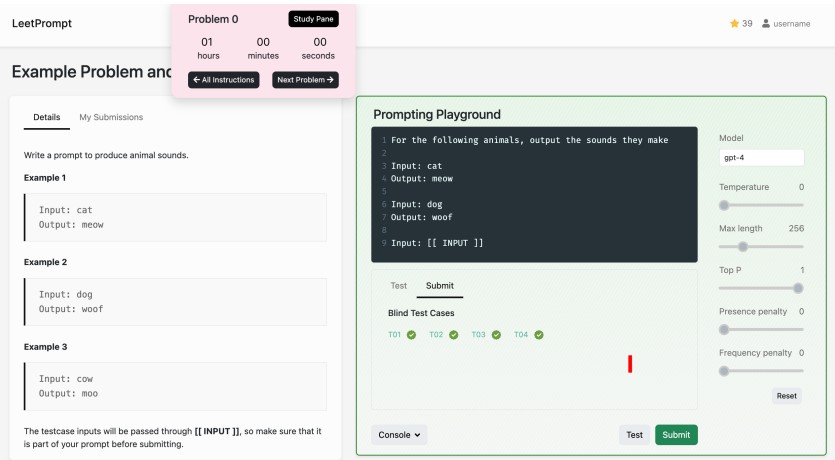

Figure 16: **Submission feedback** Once a submission is completed, participants receive feedback on how their instructions performed against the blind test cases. The feedback is shown to the participants in form of the shown the number of test cases that passed (see I). Instructions are highlighted based on number of test cases passed. Green indicates that all test cases passed, yellow for some test cases passed, and red for no test cases passed.