# OpenReview forum: "LeetPrompt: Leveraging Collective Human Intelligence to Study LLMs"
_ICLR.cc/2024/Conference — ICLR 2024 Conference Withdrawn Submission_

### Official Review · Reviewer_68mT · 2023-10-27

**Soundness:** 2 fair
**Presentation:** 3 good
**Contribution:** 1 poor
**Rating:** 5
**Confidence:** 3

**Summary:**

This paper presents "LEETPROMPT," a platform that harnesses collective human creativity to generate prompts for large language models (LLMs) and solve reasoning questions across various domains. The study involved 20 human subjects attempting 10 questions in five domains. They collected 1,173 prompts and made several observations such as: 1. system can solve previously deemed unsolvable problems, 2. participants used diverse prompting strategies, and 3. difficult questions received many prompt submissions.

**Strengths:**

First of all, the paper is well-written and the examples & figures helps the reader to understand how the system is designed, how it is used, and how it works in general.

**Weaknesses:**

My primary concern regarding this paper is its contribution and novelty within the Large Language Model (LLM) space. While I recognize the value of providing a user-friendly environment for prompt engineering, I would appreciate a clearer exploration of potential use cases for this tool within specific companies, domains, or problem-solving scenarios. Is the primary intention to create a platform for "prompt engineers" to enhance their skills? If so, as paper mentions, it appears to resemble existing platforms like Leetcode, and there is no discussion/need around representing some specific knowledge/prompt in different ways etc, that will add some novelty aspect from ICLR perspective. Acc to me, this paper can be a better candidate for a software engineering related conference.

The user base of 20 people in this study appears relatively small to support claims such as "LEETPROMPT encourages more attempts on difficult problems" or "LEETPROMPT captures diverse prompting strategies used by participants." Recognizing that finding users for a challenging task can be difficult, it would be beneficial for the paper to acknowledge this limitation.

It would be valuable to have information about the specificities of the users, such as their professions and backgrounds. This information could help in understanding how different user groups might interact with LEETPROMPT. For instance, a software engineer might find it easier to navigate the platform compared to a data analyst or someone with less experience on platforms like Leetcode. Additionally, knowledge of users' previous experiences with platforms like Leetcode could provide insights into how prior familiarity might influence their engagement with LEETPROMPT. This information could enhance the paper's contextual analysis and interpretation of the results.

Minor things:
(abstract) we conduct a study 10 questions —> including 10 questions
to upload questions, or models,in different domains —> remove commas
towards a collectively knowledge —> towards a collective knowledge
and general Knowledge —> and general knowledge
the more difficult problems received more —> received the more

**Questions:**

stated as above

---

### Official Review · Reviewer_RmkX · 2023-10-28

**Soundness:** 3 good
**Presentation:** 3 good
**Contribution:** 2 fair
**Rating:** 5
**Confidence:** 4

**Summary:**

This paper introduces a platform called LeetPrompt as a way to crowdsource prompts from humans. The authors conduct a user study (N=20) to collect prompts on 10 problems across 5 domains and compare the model’s resulting performance on human prompts against automated chain-of-thought approaches. The authors find that humans are able to create prompts that allow the model to reach perfect accuracy on the 5 hidden test cases compared to automated prompts, despite the humans themselves not being able to uniformly solve all 10 problems.

**Strengths:**

- The paper demonstrates the importance of going beyond automated prompting approaches and considering the benefits of incorporating human ideas in generating prompts.
- The paper constructs a platform that I hope can be more widely used to collect prompts and conducts a corresponding evaluation to demonstrate the benefits of human prompt generation.

**Weaknesses:**

While it seems intuitive to observe some of these takeaways that the authors state in the paper about the benefits of human prompting, but at what cost? The benefit of developing automated approaches or at least some approach that combines both automated and crowdsourced prompts is that they would be far more scalable. Some questions that come to mind are:
- How many people do you need per question? Does this vary across questions?
- What kind of onboarding is necessary for people to have a grasp of how to prompt well?
- What kind of background is required? It seems like the vast majority of these 20 participants work in academia/tech and have graduate/college/phd degrees (as shown in Figure 6). Additionally, given the anecdotal example provided in point 1 of Section 4 results, it seems that significant creativity and understanding of the domain are necessary to come up with good prompts for more difficult problems.
- What is the distribution of time spent coming up with each prompt (and writing test cases)?

While it is not a weakness that prompts collected from LeetPrompt were able to solve all hidden test cases, it is a limitation (particularly given the small number of test cases) in terms of helping the community understand what the pitfalls (if any) of collecting human generated prompts are, i.e., when are automated approaches are better or worse than humans.

**Questions:**

- What is the effect of having people only interact with GPT4 to generate prompts?
- Of the 20 participants, can you report for each question how many people actually came up with a prompt that would pass all 5 test cases (or a fraction thereof)? It would be helpful to get a sense of whether all 20 participants were contributing relatively equally or whether some participants were coming up with significantly better prompts.
- A benefit of such a platform seems like the ability to have people edit and improve on other prompts, though this was not studied in this work.
- There were multiple typos, e.g., “a study 10 questions”, should Table 6 in the text actually be Table 5?

---

### Official Review · Reviewer_EkgE · 2023-10-29

**Soundness:** 3 good
**Presentation:** 3 good
**Contribution:** 2 fair
**Rating:** 5
**Confidence:** 3

**Summary:**

This paper introduces a platform that is designed to leverage collective human creativity in constructing prompts for LLMs to solve reasoning questions across various domains. The main motivation is that while LLMs have become popular, the prompting remains challenging and often requires manual effort. This platform allows users to attempt questions by writing prompts that can solve hidden test cases. This study yielded science problems related to GPT-4 prompts and findings. This platform is a potential solution in improving high-quality data collection.

**Strengths:**

- This paper emphasises the user experience. The platform's design is likely to be familiar and intuitive to many users.
- This platform highlights the potential of collective intelligence in improving the quality of prompts.
- The idea of leveraging collective human creativity to construct prompts for LLMs is interesting.

**Weaknesses:**

- The paper introduces a platform but fails to provide a demo or code, diminishing its credibility. Additionally, the generated prompts (set) are not shared, which could have offered more transparency and understanding.
-  It seems a well-designed test case is crucial for deriving good results. The paper does not provide sufficient details on how these test cases were crafted, especially in real-world cases.
- The study's sample size is only 20 human.
- Missing related work.
- There are several existing works that offer prompts for inexperienced users. I hope the authors provide a more comprehensive comparison of these methods to clearly delineate their method‘s advantages.
- The quality of prompts can significantly influence outcomes, and any disparity might lead to unfair comparisons. When comparing with Auto-CoT, the initial prompts are very different, please give a detailed description of this process.
- This paper is more like an incremental work to me. I hope the authors elaborate the novel contributions more clearly. While the idea of crowdsourcing is interesting, its innovative angle might not align with the expectations of ICLR.

**Questions:**

- The test cases seem to have a significant impact on the final prompts/results, how does the platform ensure the quality of these test cases (in the real world)? How does it leverage collective intelligence in this process?
- If users in real-world scenarios use this platform, there remains an unresolved problem: How do either the users or the platform determine the correctness of the final results?

Suggestion:
- In Section 5, it is recommended not to use red circles as item numbers. It may lead to confuse the readers.

---

### Official Review · Reviewer_o2mR · 2023-10-31

**Soundness:** 2 fair
**Presentation:** 3 good
**Contribution:** 2 fair
**Rating:** 3
**Confidence:** 3

**Summary:**

This article proposes and designs a crowdsourcing scientific platform called leetprompt for testing large language models.

**Strengths:**

This work presents a public scientific platform for testing large-scale language models, which adopts the idea of an online programming platform. Scientists and others propose questions and provide hidden test cases, while prompt engineers adjust prompts and hyperparameters to solve the problems. The UI design of this platform is aesthetically pleasing and reasonable, with certain potential for development.

**Weaknesses:**

1. The novelty of this paper does not claim very clear. Although this work introduces a public LLMs evaluation platform, which is encouraging, I do not understand what progress this platform has made in terms of innovation. It is necessary to highlight the research motivation progress and the technique innovation in the platform.
2. For this kind of platform, resources are a more important issue. Whether these resources can truly evaluate the capabilities of large models comprehensively like HELM or BIGBench. I hope to see specific statistics and analysis on the quantity and quality of resources, user activity, and so on for this platform.
3. I haven't thought about how to have the driving force for users between platforms to voluntarily participate in this platform. In the long run, this is something that must be considered for development.
4. From my understanding, LLM will not remain completely consistent in its output when given the same input due to the presence of temperature and other hyperparameters. How does evaluation ensure consistency? Many evaluation datasets have standardized interfaces to ensure the rationality of evaluations, but it seems that this aspect has been omitted in this work.

**Questions:**

1. What research problems does this platform solve? What scientific problems does this work solve that have not been solved yet?
2. How to collect data and ensure sufficient attractiveness to maintain the healthy development of the platform.
3. How to ensure consistency and fairness of the results.
Other major concerns are listed above